Natural Hazards and Earth System Sciences Discussions

# Lightning risk assessment at a high spatial resolution using the resident sub-district scale: A case study in Beijing metropolitan areas

HaiBo Hu<sup>1</sup>, JingXiao Li<sup>2</sup>

<sup>1</sup>Institute of Urban Meteorology, CMA, Beijing, 100089, China
 <sup>2</sup>Beijing Lightning Devices Security Test Center, Beijing, 100080, China
 *Correspondence to*: HaiBo Hu (hbhu@ium.cn)

**Abstract.** Lightning risk indexes identifying the potential number of dangerous lightning events (NDLEs) and ground sensitivity to lightning in resident sub-districts of Beijing metropolitan areas have been unprecedentedly estimated on a 5 m

- resolution grid. The gridded cloud to ground (CG) lightning stroke density was used in the NDLE calculation, on account of multiple contacts formed by CG lightning flash multiplicity. Meanwhile, in the NDLE estimates, the critical CG stroke gridded densities derived from the lightning location system (LLS) data were corrected for network detection efficiency (DE). This case study on resident sub-district indicates that the site-specific sensitivity to lightning, which is determined by the terrain factors related to lightning attachment, as well as lightning rod effects induced by nearby structures, differs
- greatly across types of underlying ground areas. The discrepancy of the NDLE which is the numerical product of sensitivity and CG stroke density, is predominated by the sensitivity on account of the relatively stationary CG stroke density in a resident sub-district scale. Conclusively, the visualization of lightning risk sensitivity and NDLE discrepancy in parts of a resident sub-district at high spatial resolution makes it convenient in risk reduction and risk control for lightning risk management.

# 20 1 Introduction

The frequent occurrences of lightning disaster events cause large numbers of casualties and substantial damage losses, such that lightning is considered as one of the most dangerous natural hazards (Curran et al. 2000; Holle et al. 2005; Zhang et al. 2011) and second meteorological killer (Ashley and Gilson 2009). Lightning risk assessment is meant to investigate, search for and locate high-risk areas, enabling the implementation of mitigation measures for lightning risk reduction (Kaplan and

25 Garrick 1981; Hu et al. 2014). It is desirable to assess lightning risks at an extremely high resolution (e.g., 5 m spacing grid), sufficiently detailed in reflecting lightning risk characteristics and allowing risk discrepancies recognizable in a real-world view. This recognition makes it applicable for disaster preparedness and delivers a practical lightning risk management (Mills et al 2010).

At a high spatial resolution, it is possible of deliberately locating specific underlying areas especially in densely populated urban areas, and improving the estimates of ground sensitivity to lightning, which is correlated to certain environmental settings, such as topographical features and distribution of earthed structures (Rizk 1994; Vogt 2011). Approaches will be employed to finish the pattern recognition of topographical features, locating and determining the lightning collection areas

of earthed structures, downscaling CG stroke density into a finer grids, etc. Undoubtedly, it can be fulfilled with GIS technology supported by high resolution mappery-data.

Natural lightning strokes on the ground are obvious drivers of lighting-related disasters. Lightning climatology, preliminarily manifesting lightning risk, should be described quantitatively for risk assessment (Bogdan and Burcea 2010). The approach is to derive lightning parameters (e.g., CG flash/stroke density, CG flash multiplicity) from observational data, e.g.,

- climatological data (Changnon 1985; Gabriel and Changnon 1989), satellite sensing lightning imagery (Christian et al. 2003) and lightning location system (LLS) data (Changnon 1993; Schulz et al. 2005; Biagi et al. 2007; Cummins et al. 2009). These lightning parameters fundamentally reflect regional lightning activity relevant to lightning disaster occurrence (Schulz et al. 2005; Mäkelä et al. 2010). They are critical in confirming lightning risk even at a resident sub-districts scale. As a premise of risk recognition, lightning characteristics should be unveiled mostly by introducing LLS data, on account of its
- high spatial-temporal resolution (e.g., Krider et al. 1980). Then the lighting risk characteristics would be revealed by overlapping the lightning characteristics (CG flash/stroke density) with other risk factors (e.g., sensitivity and exposure) (Hu 2014).

Lightning risk is linked to the combined effects of regional lightning activities and ground sensitivity to lightning, reflecting on-site lightning hazards. Risk recognition at high resolution can visualize the decision-making procedures in risk

management. It facilitates the formulation of practical risk management strategies for disaster prevention (Smith 1996). For a resident sub-district, the visualized lightning risk recognition can provide information in a form that is straightforwardly understandable to local decision and policy makers. Moreover, this quantitative information about site-specific lightning risk is critical to public safety and infrastructure planning (Stallins and Rose 2008).

# 2 Data description

#### 25 2.1 Lightning location system (LLS) data

The LLS data collected from 2007-2011 by the ADTD (Advanced TOA and Direction system; TOA denotes time-of-arrival) network deployed by the China Meteorology Administration (CMA) were used to derive the CG flash/stroke density. These data include time, location, amperage and polarity of CG lightning strokes.

The ADTD consists of more than 301 sensors (by March, 2011) in China (Yao et al. 2012). Around Beijing district, nearly 9-30 14 ADTD-1 sensors, types of improved IMPACT (combined MDF and time-of-arrival (TOA)) sensors, detect 1 k-450 kHz (the very low frequency (LF) band) lightning sources (Fig. 1). The ADTD-1 sensors use the combined MDF and time-ofarrival (TOA) method for position retrieval. In this method, if lightning source is only detected by two ADTD-1 sensors, the

algorithm uses one TOA hyperbolic curve and two MDF vectors to retrieve the position; if it is detected by three sensors, in non-bilingual region, the TOA algorithm is used to retrieve the position directly, whereas the TOA is firstly used to find a bilingual locations and then using the MDF to find the true location; if it is detected by four or more sensors, a TOA least square method is used to retrieve a more precise position. So the location precision of the lightning resource reported by four

or more sensors is actually better than that reported by two or three sensors. In our LLS data, the ratio of four or more sensors reporting lightning resources to the total is 66.815%.

The DE of ADTD sensors is claimed to be 90% in 300 km, with a 600 km maximum detectable distance, and a location accuracy error within 1 km. However, 90% of the flash DE can be validated, but with a lower stroke detection efficiency (SDE). The first stroke peak current in a multiplicity CG flash can be greater than twice of its subsequent stroke peak current

(Rakov and Uman 1990); thus, the sensors can capture the first larger peak stroke but missing its weak subsequent (Rudlosky and Fuelberg 2010). Moreover, some weak CG strokes (including a single-stroke CG flash) cannot be detected due to signal attenuation induced by long distance propagation and terrain factors (Sch ütte et al. 1988), etc., .
The stroke number is critical in lightning risk estimates (Bertram and Mayr 2004). Thus, we estimated the SDEs of the

ADTD in grids (1 km×1 km size, see Fig. 1) around Beijing and corrected the lightning stroke density for network DE. The

15 SDE estimates approximate those of the U.S. NLDN (National Lightning Detection Network) in 1998, which was reported to be 62% (Idone et al. 1998). So the DE level of ADTD is equivalent to that of the NLDN at least in 1998, indicating a great improvement left for network upgrades.

#### 2.2 Others

Digital elevation model (DEM) data were used to identify site-specific lightning attachment capabilities on account of the 20 topography (Vogt 2011). Its 30 m spatial resolution basically meets the need of identifying hypsographic features and confirming terrain factors.

Additionally, the basic GIS data with map scales of 1:2000 in urban settings and 1:50,000 in rural settings have been used to measure a structure's lighting collection area, incorporating the structure's geometric shape and height, which are readily available in GIS map layers (Hu et al. 2014). The dataset of GIS map-layer keeps the structure-type field which can be used

to determine the structure lightning protection capability.

### **3 Methods**

The lightning risk index of the potential number of dangerous lightning events (NDLE) is preserved for lightning risk zoning at a resident sub-district scale. Correlated to regional lighting activity and sensitivity to lightning at a site, the NDLE  $N_x$  can generally be estimated as (Hu et al. 2014)

$$30 \quad N_x = K \times N_g \times A_d \,, \tag{1}$$

where K denotes the coefficient related to environmental settings onsite;  $N_g$  the CG lightning stroke density (stroke/yr.km<sup>2</sup>); and  $A_d$  the collection area of the lightning strike, mostly determined by site-specific lightning attractiveness variably in types of underlying ground areas. On account of each stroke in a multiple-stroke CG flash can produce damage losses and/or casualties, it is reasonable of taking  $N_g$  to be the CG stroke density (stroke/yr.km<sup>2</sup>).

#### 3.1 CG lightning stroke density corrected for DEs and downscaling 5

The CG stroke density  $N'_{g}$  derived from LLS data was corrected for DEs of the ADTD using the following equation.

$$N_g = \frac{N'_g}{D_g},\tag{2}$$

where  $N_g$  is the corrected CG stroke density;  $D_g$  the DE in grids.

The CG stroke density in the 5m grids had been downscaled from 1 km grids. We used the approach of inverse distance 10 weighting (IDW) to interpolate the CG stroke density from these of the larger grids (1 km spaced), which 9 grid cells were involved, including the containing, up, down, up-left, up-right, left, right, down-left, and down-right 1km×1km grid-cell. Mathematically, the interpolation can be described as

$$N_{g5m} = \sum_{i=1}^{n} N_{g}(i) \cdot \frac{1/r(i)}{\sum_{n=1}^{n} 1/r(i)},$$
(3)

where  $N_{g5m}$  is the interpolated CG stroke density in 5m spaced gridcell;  $n (n \le 9)$  the number of the containing and its around 1km spaced grid-cells;  $N_{g}(i)$  the CG stroke density of the *i*th 1km spaced grid-cell; r(i) the distance of the center point 15 of the 5m gridcell to that of the *i*th 1km spaced grid-cell.

#### 3.2 NDLE estimates in 5 m spacing grids

We calculated the NDLEs on earthed structure, outdoor area under a structure canopy and an open-field area, respectively, on account of difference in estimating their lightning protection capability, lightning attachment and lightning attractiveness.

20 The approaches are correspondently adjusted in conditions of the grids intersecting on different types of underlying areas (Fig. 2).

#### 3.2.1 NDLE estimates of an earthed structure (ES)

The NDLEs of a structure  $N_d$  is calculated as (Hu et al. 2014):

$$N_d = N_e A_d C_d P_d . 10^{-6}, (4)$$

where  $A_d$  (m<sup>2</sup>) is the collection area of a structure to lightning;  $C_d$  the terrain factor, which is deduced using DEM data, 25 accounting for its relationship to the surrounding topography (see Table1);  $P_d$  is the coefficient representing the lightning protection capability of the structure.

Given the structure height in meters H, the collection area  $A_d$  can be determined as (Rizk 1994)

 $A_d = 670.8\pi H^{0.96},$ 

(5)

The structure protection capability includes these of protecting 1) the live beings from injured by a lightning stroke, 2) the structure from physical damage, and 3) the internal systems in the structure. Substantially, these capabilities are represented

- by the casualty probability  $p_a$ , the physical damage probability  $p_b$ , and the internal systems failure probability  $p_c$ , respectively in risk estimates. Herein, for simplification, only  $p_a$  is taken into account of the lightning risk assessment, i.e.,  $P_d = p_a$ . The casualty probability due to touch and step voltage induced by a lightning stroke to the structure, reflects the structure Lightning Protection Level (LPL), which can be determined according to the lightning protection measures taken by a structure (Table 2).
- We have defined the protection measures that would be probably taken by 10 structure types in Beijing (see Table 2), readable in a GIS map-layer dataset. Most structures are equipped with lightning rods. Some concrete steel structures have iron infra-structure and framework as the lead-in wire for lightning protection. Thus they possess a better capability of protecting the live beings from injured by ground lightning stroke.

# 3.2.2 NDLE estimates of an outdoor area under a structure canopy (OAUSC)

Under this condition, the NDLEs  $N_{dc}$  can be calculated as

$$N_{Dc} = N_g A_{Dc} C_d C_c \cdot 10^{-6}, (7)$$

where  $A_{Dc}$  (m<sup>2</sup>) is the intersection area of the OAUSC and the grid cell;  $C_d$  the terrain factor of the grid cell; and  $C_c$  the coefficient representing lightning rod effects produced by the surrounding structures. At a fine grid scale (e.g., 5 m), its calculation is simplified as follows (Petrov and D' Alessandro 2002):

$$C_c = \frac{1}{\sum_{i=1}^{n} H(i)}$$
, (8)

where H(1), ..., H(n) are the floor numbers of the surrounding structures, whose canopies cover the grid cell. It is conceivable that  $C_c$  will approximate zero if the grid cell is under canopies of many nearby tall structures.

### 3.2.2 NDLE estimates of an outdoor area under a structure canopy (OAUSC)

Totally exposed to lightning stroke, open-field area is more susceptible to lightning. Thus, its NDLEs  $N_{Ds}$  can be estimated 25 as

$$N_{Ds} = N_g \cdot A_{Ds} \cdot C_d \cdot 10^{-6}, (9)$$

where  $A_{Ds}$  is the intersection area of the OFA and the grid cell.

(10)

(11)

#### 3.2.2 NDLE estimates of an outdoor area under a structure canopy (OAUSC)

After the NDLEs of the three types of underlying ground areas are calculated out, the NDLEs of a grid cell intersecting with these areas, *Nd\_Cell*, can be calculated as

$$\begin{split} Nd \_ Cell &= N_d \times Inter \sec t(Area \_ Cell, Area \_ ES) \\ &+ N_{Dc} \times Inter \sec t(Area \_ Cell, Area \_ OAUSC) \\ &+ N_{Ds} \times Inter \sec t(Area \_ Cell, Area \_ OFA) \end{split}$$

where *Area\_Cell, Area\_ES, Area\_OAUSC,* and *Area\_OFA* denote the geometries of the grid cell, earthed structure, outdoor area under structure canopy and open-field area in the grid cell, respectively, and *Intersect* is a GIS operator of calculating the intersection areas of the grid cell and the geometries of the three types of underlying ground areas (i.e., the structure, outdoor area under structure canopy and open-field area), respectively.

#### 3.3 Parameters reflecting lightning risk characteristic

The lightning risk assessment follows the workflow of 1) estimating NDLEs and sensitivity, 2) searching out high risk areas, and then 3) providing pertinent advice for decision makers who will take measures addressing lightning risk mitigation in the resident sub-district.

The CG stroke density,  $N_g$ , ground sensitivity to lightning,  $S_x$ , and NDLEs,  $N_d$ , essentially reflect the lightning risk characteristics in a local community with respect to decision making in lightning risk management. The CG stroke density,

- $N_g$ , which is an indicator of regional lightning activity, can be derived from the LLS data. The NDLEs, a numerical product of the CG stroke density  $N_g$  and sensitivity  $S_x$ , reflect the lightning hazardousness at a site. In definition, the sensitivity is an indicator of proneness to lightning strike, comprehensively measured by underlying ground lightning attractiveness, lightning protection, and lightning attachment, correlated to land-surface characteristics, e.g., terrain features and existence of earthed structures. Accounting for site-specific environmental settings rather than regional lightning activity, it can be 20 calculated as
  - $S_{d} = A_{d} \bullet C_{d} \bullet P \bullet 10^{-6},$

$$A_{Dc} = A_{Dc} \bullet C_d \bullet 10^{-6}, \tag{12}$$

$$S_{DS} = A_{DS} \bullet C_d \bullet 10^{-6}, \tag{13}$$

where  $S_{dr} S_{Dcr}$  and  $S_{Ds}$  denote the sensitivity to lightning on a structure, outdoor area under a structure canopy and open-field 25 area, respectively. The discrepancy of NDLEs in a sub-district is mostly determined by that of the sensitivity, due to the relatively stationary CG stroke density. In this context, the sensitivity and the NDLEs jointly describe the lightning risk characteristics at a high resolution.

## 4 Analysis on lightning characteristics

Lightning climatology preliminarily reflects lightning risk, not accounting for the sensitivity and exposure to lightning (Ashley and Gilson 2009). Analysis on lightning characteristics is the premise of risk assessment even in a sub-district, at lest it can provide the critical parameters for lightning risk assessment, e.g., the CG flash/stroke density and CG multiplicity.

We derived the lightning parameters from the ADTD data by counting the annual CG flash/stroke numbers at a resolution of 1 km. The CG strokes were grouped into flashes based on a multiplicity delay of 1 s within a radius of 20 km (Cummins et al., 2006, Drüe et al. 2007) and +CG flashes with a peak current of less than 15 kA were classified as IC lightning (recommended by Cummins and Murphy 2009).

Convection events are usually enhanced by orographic uplift in the mountains, which trigger more CG strikes (Bourscheidt

- et al. 2009). However, the derivation from the ADTD data exhibits a relatively lower CG flash/stroke densities in the north and west mountainous areas than that in the plains, except for a relative high density in the south-west mountains (Fig. 3). Cummins et al. (2006) suggested that an elevated terrain and conductivities of the underlying surface have a stronger influence on the attenuation of the signal produced by CG flashes, in turn reducing the network DE. Moreover, the thunderstorms in urban areas on the plains can be enhanced by urban characteristics (e.g., roughness, aerosols, and urban
- heat islands) and consequently induce more CG flashes (Shepherd et al. 2002; Rose et al. 2008; Stallins and Rose 2008; Hu et al 2014, 2015; Kar and Liou 2014), e.g., the downwind areas having high CG stroke densities (see the blue-circled in Fig 3.c). Also, a high CG stroke density distributed in upwind southern areas (see the purple-circled in Fig 3.c) can be perceived. We assumed that it should be related to random cloud condensation nuclei (CCN) concentrations affecting the clouds over cities (Steiger et al. 2002; Stallins et al. 2006; Kar and Liou 2014).
- No matter what can explains the higher CG flash/stroke density in the plains, the DE of a LLS cannot be 100% (Schulz et al. 2005; Mazarakis et al. 2008). The actual CG stroke numbers in the grids, however, are required in the NDLE estimates. Thus, we corrected the grided CG stroke densities for DEs to fit the actual.

Network DE is determined by the performance and sensitivity of sensors, the sensor network geometry, and the underlying ground conductivity (Schütte et al. 1988; Naccarato and Pinto 2009; Mäkelä et al. 2010), etc., . The capability of DE

- estimates in correcting CG flashes/strokes and evaluating the LLS network performance invoked a series of methodologies published in the literature (e.g., Schütte et al. 1988; Cummins et al. 1998; Naccarato and Pinto 2009). Although DE can be determined more precisely with observations collecting exact information of lightning occurrences (e.g., video or tower measurements), this approach can only be experimentally utilized in producing localized DE estimates (Saraiva et al. 2010; Visacro et al. 2010; Warner et al. 2013). The methods of DE estimates using theoretical models are more convenient and
- applicable in comprehensively confirming a network DE. Schütte et al. (1987, 1988) introduced the Weibull-distribution into sensors' signal strength acceptance estimates, facilitating network DE calculations. Cummins et al. (1998) also combined the peak current cumulative distribution with a signal-propagating model to estimate the absolute flash DE for the NLDN.

/

10

20

Naccarato and Pinto (2009) deduced the DE values using the sensor's individual DE probability functions derived from a large network detected CG stroke data, considering different distances from the sensors and specific peak current ranges. We calculated the DEs of the ADTD in raster grids according to the networks performance and sensitivity measured by the distances and azimuths among sensors.

5 Methodologically, in case of the weibull distribution of signal strength (Schutte et al. 1987), the signal acceptance of a sensor can be given by

$$A(r) = \begin{cases} 0 & r < cr_0 / s_{\max} \\ 1 - \exp\left[-\left(\frac{s_{\max}r/r_0 - c}{a}\right)^b\right] & cr_0 / s_{\max} < r \le cr_0 / s_{\min} \\ \exp\left[-\left(\frac{s_{\min}r/r_0 - c}{a}\right)^b\right] - \exp\left[-\left(\frac{s_{\max}r/r_0 - c}{a}\right)^b\right] & r > cr_0 / s_{\min} \end{cases}$$
(14)

where  $s_{min}$  and  $s_{max}$  are the lower and upper signal threshold, which will be 20 and 600 arbitrary units (a. u.), respectively;  $r_0$  the standard distance, which will be 100 km; *r* the distance to the sensor; *a*, *b*, and *c* are the scale, the shape and the location parameter of the Weibull distribution of signal strength (Sch ütte et al. 1987, 1988).

Only two ADTD IMPACT sensors reporting a stroke are required to get a valid solution. Thus, the DE on a grid-cell can be determined as (Naccarato and Pinto 2009)

$$A = A_1(r_1) \times A_2(r_2) \quad A_1 ranked \qquad A_1(r_1) \ge A_2(r_2) \ge A_3(r_3) \ge \dots$$
(15)

where  $A_i(r_i)$  denotes the acceptance of one sensor;  $r_i$  (i=1,2,3,...) is the distance of the *i*th nearest sensor to the grid-cell 15 center and *A* the grid-cell DE of the network.

After corrected using these deduced DEs (see Fig. 1), the CG stroke densities in the northeast mountains, metropolitan areas, south plains and southwest mountains increased significantly in comparison with the uncorrected CG stroke densities (see Fig. 3b-c). The corrected densities in metropolitan areas are mainly between 4-6 stroke/yr.km<sup>2</sup>, which is higher than expected. However, the relatively high CG stroke density remains in the plains. It is advisable that the network should be upgraded for improvement of the network DE and detection accuracy. Maybe this anomaly can be explained using observed

evidence.

### 5 Case study of lightning risk assessment in a resident sub-district

The model running out at a 5m resolution optimally covers a small area of 10-100 km<sup>2</sup>. We selected two resident subdistricts in Beijing metropolitan areas for risk analysis on the indicators of sensitivity and NDLEs. One selected is the sub-

25 district of Malianwa located in the northwest metropolitan areas and foothills of the western YanShan Range. Its complex topography implies a diversity of ground sensitivity to lightning. The other is Beijing International Airport, where the lightning risk discrepency is remarkable between the open fields of the aircraft parking areas and the terminal structure inside.

### 5.1 Ground sensitivity to lightning

Sensitivity recognition contributes to lighting risk avoidance on thunderstorm days, in respect to risk management. Also, it can be used in directing deployment of lightning protection facilities and systems (Schulz et al. 2005).

- The lightning sensitivity zoning in the sub-district of Malianwa indicates that the sensitivities of structures and outdoor areas under structure canopies are usually less than 0.15 in magnitude (Fig. 4a). Alternatively, if not accounting for the terrain factors, the greatest sensitivity is 1.0 on an open field in the plains (Fig. 4a). Accounting for the terrain factors, the sensitivity in mountainous areas will increase to 1.15-1.3, which occurs, for example, in the western uplands of this sub-district (see *A* in Fig. 4b-c), where the high sensitivity zones spread. This higher sensitivity in the hills means that the CG strikes would favor topographic highpoints by as much as 15-30.0% when compared with random points in the plains. This increased sensitivity of topographic highpoints is somewhat in agreement with the findings of Vogt (2011).
- Displayed in Google Earth, the sensitivity zones exhibit a good correlation with topographical features and distribution of earthed structures (see Fig. 4c). Apparently, the simulated sensitivity is explicably in accordance with the settings and it merits visualization in lightning risk management.

### 5.2 NDLEs

Similar to sensitivity, NDLEs on a structure and an outdoor area under structure canopy are lower, while the NDLEs on an open-field area are equal or even magnitudes greater than the CG stroke densities of the downscaled grids. NDLEs of the uplands in western Malianwa exhibit this pattern where more upward and/or downward lightning can be triggered on account of topographic highpoint attachment (Warner et al. 2013).

The advantage of quantitative risk assessment at high resolution is that its visualized risk characteristics can play an

- important role in operating risk control effectively. For instance, at Beijing International Airport, terminal 3 (a 45 meter high structure) and its nearby outdoor areas under structure canopies exhibit a low assessed sensitivity of 0.15, equivalent to 0.15 times that of an open-field area, and NDLEs below 0.5 (time/yr.km<sup>2</sup>) (Fig. 6). Conversely, the red-ellipse in the aircraft parking apron, hundreds of meters away from the terminal, exhibit a high sensitivity of 1.0, and NDLEs above 2.5 (time/yr.km<sup>2</sup>) (Fig. 6), due to the lack of lightning protection and structure shelter. On Aug. 11th, 2013, a lightning fatality
- occurred exactly within the red-ellipse (Fig. 6b), wherein a cleaning staff member was struck dead by lightning when using a mobile phone (Hu 2014). Therefore, the personnel should pay attention to lightning on thunderstorm days when operating in open fields. Moreover, lightning risk management should be conducted based upon risk recognition in the airport community so that it can visually inform where is safe or not (Fig. 6).

#### 6. Conclusion

The DE of a lightning location system (LLS) cannot be 100% and low DE are usually due to lack of deployed network sensors, as well as the performance and sensitivity of sensors. Meanwhile, the signals produced by CG flashes can be

strongly attenuated by long distance propagation, terrain factors and underlying conductivity. Before being used in NDLE estimates, the CG stroke densities derived from LLS data could be corrected for DEs. Although the correction of CG stroke density makes it better qualified for risk assessment, it is preferable that the LLS data should be more reliable with the network upgrading to improve the DE and location accuracy (Rudlosky and Fuelberg 2010). Moreover, network upgrades should be implemented not only for optimal lightning location in the metropolitan areas but also in the mountainous rural

- areas, where more lighting casualties occur (López and Holle 1998; Curran et al. 2000; Zhang et al. 2011). The uncertainty of lightning risk estimates at this high resolution will be induced by the LLS data quality related to location precision, and imperfect network DEs. Also, the model structures and operations (e.g., CG stroke density downscale) magnify this uncertainty. Though the IDW interpolation and the overlapping of derived CG stroke density with the ground
- sensitivity to lightning somehow attenuate the errors of risk estimate, the uncertainty remains. However, it can be suggested the uncertainty caused by LLS data quality should be reduced by network upgrades with adding and/or replacing a higher performance and sensitive sensors. Moreover, further research can be undertaken to evaluate the reliability of this risk estimates in uncertainty (e.g., Monte Carlo simulation), or even to find an effective approach of uncertainty reduction leading to more precise calibration and correction.
- The model running at a fine resolution (e.g., 5m grids) enables lightning risk assessment accessibly accounting for the ground sensitivity of the types of underlying ground areas, as well as overlapped with CG stroke density. The lightning risk recognition at high resolution can reveal risk discrepancies visually and locate higher risk areas at a finer scale, making it favorable in lightning risk management.

This case study indicates that the lightning rod effects of structures make low risk in an outdoor area under its canopy. In

comparison, an open field area usually exhibits a higher risk, with its NLDE equal to the corresponding CG stroke density and its sensitivity nearly 1.0 in magnitude. On account of terrain factors, NLDEs and sensitivity can increase by 1.15-1.3 times in uplands vs. the plains due to higher lightning attachment in elevated areas.

The distributions of lightning parameters (e.g., CG flash/stroke density), ground sensitivity to lightning and NDLE comprehensively reveal lightning risk characteristics. The CG lightning flash/stroke density, CG flash multiplicity, etc.,

- mostly derived from the LLS data, not only indicate the regional lightning activity but also constitute the input parameters for lightning risk assessment. The sensitivity correlates to site-specific lightning attractiveness, lightning protection capability and lightning attachment. These qualities are determined by the site conditions, including the existence of structures and the topography of the site. The sensitivity indicates which parts of a resident sub-district are relatively prone to lightning strike. The NDLE reflects lightning hazardousness, accounting for both regional lightning activity and sensitivity.
- The CG stroke density, sensitivity and NLDEs are indicators critical for decision making in risk reduction, response to taking effective actions, e.g., erecting warning boards in high risk areas, installing lightning protection facilities in the domains susceptible to lightning, or even constructing a temporary structure serving as thunderstorm shelter in a public open field area. All will attain the goals of lightning risk management in a resident sub-district.

# Acknowledgements

This study has been supported by the National Natural Science Foundation of China (Project: 41175099) and the Beijing Natural Science Foundation of China (Project: 8142019).

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
