# Peer review of "Lightning risk assessment at a high spatial resolution using the resident sub-district scale: A case study in Beijing metropolitan areas"

_Natural Hazards and Earth System Sciences, 2016_

## Referee Comment (RC1) · Anonymous Referee #1 · 22 Sep 2016

Lightning risk assessment . . . by Hu and Li

The paper presents interesting arguments concerning risks from lightning strokes in areas of varying terrain features. After some revisions it can be published. The parts that relate to technical risks and the corresponding quantitative considerations are sound and will not be criticized. However, the handling of lightning needs substantial changes as is explained below. 1. The paper deals with CG strokes. It is not explained why IC strokes are not present, or how they are eliminated. That can disturb the CG density estimate. The strange procedure to declare that CG+ are in fact IC+ is not sufficient or satisfactory. 2. The authors mention multiple contacts of strokes that occur within one flash. However, there are also multiple strokes that contact ground at the same point.

When one deals with strokes densities (Ng), this point becomes crucial. 3. A location accuracy of 1 km is claimed. This is much too optimistic when 1/3 of the locatings are performed with only 2 or 3 sensors. DF is known to produce errors of many km. 4. On p.5 line 8 the authors claim that 90% of the flash DE can be validated. This statement is not understandable and needs a clear explanation. 5. On p.5 line 7 it is claimed that the sensor range is 300 or 600 km. This is not correct, because the detection depends on the current of the stroke. A weak current of 4 kA will not be detected at all, while a 100 kA stroke can be seen as far as 1000 km. 6. It is absolutely necessary to show a current distribution of the used strokes. 7. It is not understandable how the authors "calculate" or estimate the true DE of the system (result in Fig. 1). The true occurrence of strokes is not known; in particular, the true current distribution is unknown. Thus, there is no way to determine the absolute DE. The peak of the current distribution is the only parameter that allows estimate of relative DE with respect to other networks. It may be suspected that the CMA network exhibits the peak above 10 kA; then, the DE would be quite low because strokes with current around 5 kA are very prominent, as can be seen from highly sensitive networks elsewhere. All together, the DE scaling is not convincing and could be replaced by a mere guess. 8. On p. 7 line 13 the authors speculate that the stroke signals can be absorbed by terrain effects such that the stroke density decreases. This is not in accordance with solid observations that propagation is very well approximated by 1/D (distance D). 9. It is understandable that technical structures (different buildings) give rise to different stroke risks. Insofar, a small grid size is meaningful, although 5 m is much too small. For lightning risks, however, a grid size of 5x5 m is totally unacceptable. The authors should consult the international norm for determining flash densities (IEC 62858). Both the rare occurrence (in 5x5 $m^2$ the stroke chance is 1 stroke every 10,000 years), and the large location error of the network prohibit such a procedure.

---

## Referee Comment (RC2) · Anonymous Referee #2 · 13 Oct 2016

The authors present a lightning risk assessment of different structures but do not make it clear what the improvement/difference to the lightning risk assessment in the standard IEC 62305-2 is.

Further they use really strange assumptions, e.g. - They use the stroke densities for risk calculation. To really estimate the lightning risk the ground strike point density should be used (see IEC 62858). - The grid size of the stroke densities is not clear and it is further not clear if the amount of data is enough to calculate a meaningful lightning density with an accuracy of +/- 20%. For such an accuracy about 80 strikes should be within a grid cell (Diendorfer, 2008, and also IEC 62858). It is also not clear why the authors use 5m resolution for risk calculation. In my opinion this does not make

sense for lightning risk calculation. The lightning location system (LLS) accuracy is not even close to that value (for really high performing LLS in the low frequency range the median accuracy is in the range of ∼100m).

The authors mention that LLS data was corrected for detection efficiency but do not tell how it was corrected or with which method.

The paper is compiled very superficial with a lot of small mistakes. E.g. chapter 3.2.2 exists three times with the same chapter heading and slightly different content.

References: Diendorfer, G. (2008). Some Comments on the Achievable Accuracy of Local Ground Flash Density Values. In Lightning Protection (ICLP), 2008 International Conference on (pp. 1–6). in proceedings, Uppsala, Sweden.

---

## Author Comment (AC1) · 21 Oct 2016

The authors present a lightning risk assessment of different structures but do not make it clear what the improvement/difference to the lightning risk assessment in the standard IEC 62305-2 is. Response: We admitted the fundamentals of our risk assessment still follow the standard IEC 62305-2t. However, some work had been done on the improvement of lightning risk assessment. The primary improvement is that the resolution is downscaled from 1 km spacing grids to that of 5m spacing grids. Thus, the risk recognition in this high resolution sufficiently detailed in reflecting lightning risk characteristics and allowing risk discrepancies recognizable in a real-world view. Technically, the newly methods of CG lightning stroke estimates, downscaling, and correction for

detection efficiency, as well as NDLE estimates in 5m spacing grids are different from that in 1km spacing grids. However, we revised our manuscript slightly on emphasizing on the improvement/difference. Please see p4. line23-26.

Further they use really strange assumptions, e.g. - They use the stroke densities for risk calculation. To really estimate the lightning risk, the ground strike point density should be used (see IEC 62858). Response: The stroke density is derived from the observed data and the ground strike point density is determined based on a multiplication factor of 2 on flash density (IEC 62858). So it is puzzling to tell which is better than other. Thus, we accepted the suggestion of referee #2, and added the distribution of strike point density (NSG=2*Ng) and calculated the NDLE using strike point density suggested by IEC 62858. Please see Fig. 4d, Fig.6c, p.4 line 5-6, and p. 10 line 7-9.

- The grid size of the stroke densities is not clear and it is further not clear if the amount of data is enough to calculate a meaningful lightning density with an accuracy of +/- 20%. For such an accuracy about 80 strikes should be within a grid cell (Diendorfer, 2008, and also IEC 62858). It is also not clear why the authors use 5m resolution for risk calculation. In my opinion this does not make sense for lightning risk calculation.

Response: Our purpose is to assess the lightning risk in an extremely high resolution, which enables visualization of the relative low and high risk areas in real world view, critical for disaster preparedness and practical lightning risk management. Technically, based on an interpolation method, the CG stroke densities of 1*1 km grids were downscaled to these of 5*5 m. However, this procedure kept the original uncertainty of derived CG stroke densities in 1*1 km grids, which in some degree (not fully) are in accordance with the grid size requirement of IEC62858 for obtaining an uncertainty of less than 20% and 90% confidence level. Moreover, to meet the requirement of IEC62858, we appended 2012-2014 ADTD data, meaning recently 8 years of LLS data (2007-2014), were used to derive the CG flash density, stroke density and strike-point density, which is critical to NDLE estimates. So we replaced figure 4 and 6, and edited the corresponding paragraphs. Please see Fig. 4, Fig. 6, p. 2 line 24, and p.8

line 23.

The lightning location system (LLS) accuracy is not even close to that value (for really high performing LLS in the low frequency range the median accuracy is in the range of 100m). Response: We had corrected the paragraph about the lightning location system accuracy. However, the ADTD parameter of median location accuracy within 1 km is provided by the manufacturer. Please see p.3 line 7.

The authors mention that LLS data was corrected for detection efficiency but do not tell how it was corrected or with which method. Response: We estimated the detection efficiency using the method in reference to that of Schütte et al. (1988), and Naccarato and Pinto (2009). We added some description of our DE estimate method for a clear introduction of its fundamentals. Please see p.8 line 6-9 and line 20. .

The paper is compiled very superficial with a lot of small mistakes. E.g. chapter 3.2.2 exists three times with the same chapter heading and slightly different content. Response: We had corrected it. Please see P.6 line 1 and P.6 line 6.

Reference

Tyahla, L. J., and R. E. Lo'pez, 1994: Effect of surface conductivity on the peak magnetic field radiated by first return strokes in cloud-to-ground lightning. J. Geophys. Res., 99 (D5), 10 517–10 525. IEC62858. Lightning density based on lightning location systems (LLS)-General principles. IEC, Geneva, Switzerland, 2015.

Please also note the supplement to this comment:
http://www.nat-hazards-earth-syst-sci-discuss.net/nhess-2016-231/nhess-2016-231-AC1-supplement.pdf
* * *

---

## Author Comment (AC2) · 21 Oct 2016

The paper presents interesting arguments concerning risks from lightning strokes in areas of varying terrain features. After some revisions it can be published. The parts that relate to technical risks and the corresponding quantitative considerations are sound and will not be criticized. However, the handling of lightning needs substantial changes as is explained below.

1. The paper deals with CG strokes. It is not explained why IC strokes are not present, or how they are eliminated. That can disturb the CG density estimate. The strange procedure to declare that CG+ are in fact IC+ is not sufficient or satisfactory. Response: Drüe et al. (2007) concluded that IC-detection efficiency of ADTD LPATS sensor is

[Figure]

very low, estimated to be around 1%. The ADTD can only determine the latitude and longitude of CG lightning and cannot efficiently locate the IC lightning, differing from SAFIR which should deal with IC strokes. Thus the LLS data of ADTD are only related to CG lightning so we do no work on IC strokes.

2. The authors mention multiple contacts of strokes that occur within one flash. However, there are also multiple strokes that contact ground at the same point. When one deals with strokes densities (Ng), this point becomes crucial. Response: Although multiple strokes can contact ground at the same point, they produce multiple strokes on the ground and should be recounted multiply on account of being considered as causing disaster factor. Meanwhile, using the LLS data, it is impossible of determining the actual ground strike-point density. So, following the standards of IEC 62858, we supplementarily derived the ground strike-point density from the LLS data and used it in lightning risk estimates. Please see Fig 4d, Fig 6c, p.4 line 5-6, p. 8 line 26-27, and p.10 line 7-9.

3. A location accuracy of 1 km is claimed. This is much too optimistic when 1/3 of the locatings are performed with only 2 or 3 sensors. DF is known to produce errors of many km. Response: Admittedly, our description in p.3 line 7 is imprecise and we reedited it to be "The manufacturers claimed that the DE of ADTD sensors could be 90% in a distance between 300 and 600 km, with a median location accuracy within 1 km". Please see p.3 line 7.

4. On p.5 line 8 the authors claim that 90% of the flash DE can be validated. This statement is not understandable and needs a clear explanation. Response: We reedited the statement to be "However, only the flash DE can be 90%, but with a lower stroke detection efficiency (SDE)". Please see p.3 line 8.

5. On p.5 line 7 it is claimed that the sensor range is 300 or 600 km. This is not correct, because the detection depends on the current of the stroke. A weak current of 4 kA will not be detected at all, while a 100 kA stroke can be seen as far as 1000 km. Response:

Interactive
comment

[Figure]

We reedited it. Please see p.3 line 7.

6. It is absolutely necessary to show a current distribution of the used strokes. Response: We accordingly appended the histograms of probability density of +CG, –CG and total lightning peak currents, respectively. Please see the newly-added Fig. 1 and p. 3 line 5-6.

7. It is not understandable how the authors "calculate" or estimate the true DE of the system (result in Fig. 1). The true occurrence of strokes is not known; in particular, the true current distribution is unknown. Thus, there is no way to determine the absolute DE. The peak of the current distribution is the only parameter that allows estimate of relative DE with respect to other networks. It may be suspected that the CMA network exhibits the peak above 10 kA; then, the DE would be quite low because strokes with current around 5 kA are very prominent, as can be seen from highly sensitive networks elsewhere. All together, the DE scaling is not convincing and could be replaced by a mere guess. Response: We added some description of our DE estimate method for a clear introduction of its fundamentals. Please see p.8 line 6-9 and line 20.

8. On p. 7 line 13 the authors speculate that the stroke signals can be absorbed by terrain effects such that the stroke density decreases. This is not in accordance with solid observations that propagation is very well approximated by 1/D (distance D). Response: We give the statement referring to the literatures of Cummins et al. (2006) et al. and Schütte (1988), which declared a damping factor in different underlying surface conductivities. However, Tyahla and Lo'pez (1994) concluded that the conductivity of the underlying surface does not significantly affect the magnitude of the peak magnetic field, and hence, the peak current, in the first return stroke of a cloud-to-ground lightning flash. At this point, it is controversial of explaining the relative low CG stroke density in maintains. Accepting the referee's advice, we delete the statement. Please see p.7 line 16.

9. It is understandable that technical structures (different buildings) give rise to different

stroke risks. Insofar, a small grid size is meaningful, although 5 m is much too small. For lightning risks, however, a grid size of 5x5 m is totally unacceptable. The authors should consult the international norm for determining flash densities (IEC 62858). Both the rare occurrence (in 5x5 m2 the stroke chance is 1 stroke every 10,000 years), and the large location error of the network prohibit such a procedure. Response: The resolution of lightning risk assessment downscaled from 1km grid size of CG flash density to that of 5m just demonstrated that, in the framework, it is possible of fulfilling the extremely high resolution risk assessment, which is presented as a case study in our manuscript. The model results in this high resolution not only give the lightning risk of NDLE in each grid cell, representing risk at specific sites, but also the distribution of NDLE in a region of resident sub-district. It presents relative high and low lightning risk zonings in that region, contrasting with the terrain characteristics of underlying surface and distribution of man-made structures, and even visualizing lightning risk characteristics in real world. Then risk recognition at this high resolution facilitates the formulation of practical risk management strategies for disaster prevention and provides information in a form that is straightforwardly understandable to local decision and policy makers. Technically, based on an interpolation method, the CG stroke densities of 1*1 km grids were downscaled to these of 5*5 m, essentially having a very little chance of being stricken by lightning on account of its small exposed area. However, this procedure kept the original uncertainty of derived CG stroke densities in 1*1 km grids, which in some degree (not fully) are in accordance with the grid size requirement of IEC62858 for obtaining an uncertainty of less than 20% and 90% confidence level. Furthermore, to meet the requirement of IEC62858, we appended 2012-2014 ADTD data, meaning recently 8 years of LLS data (2007-2014), by now, were used to derive the CG flash density, stroke density and strike-point density. So we replaced figure 4 and 6b, and edited the corresponding paragraphs. Please see Fig. 4, Fig. 6, p. 2 line 24, and p.8 line 23.

* * *
[Figure]

**Supplement:**

**Lightning risk assessment at a high spatial resolution using the resident sub-district scale: A case study in Beijing metropolitan areas**

HaiBo Hu[1], JingXiao Li[2]

[1]Institute of Urban Meteorology, CMA, Beijing, 100089, China
5 [2]Beijing Lightning Devices Security Test Center, Beijing, 100080, China

*Correspondence to*: HaiBo Hu (hbhu@ium.cn)

[revised manuscript text omitted]

**3.2 NDLE estimates in 5 m spacing grids**

We calculated the NDLEs on earthed structure, outdoor area under a structure canopy and an open-field area, respectively, on account of difference in estimating their lightning protection capability, lightning attachment and lightning attractiveness. The approaches are correspondently adjusted in conditions of the grids intersecting on different types of underlying areas (Fig. 23). The spatial relationship of underlying areas to the grids illustrates that one grid box often covers only one unique underlying area and seldom covers multiple types of areas (Fig. 3). Thus, the NDLE of each grid cell can be calculated directly using a GIS overlapping operator. This approach is different from that used for 1×1 km size grids, which is to sum the total NDLEs of all areas of the big grid (Hu et al. 2014).

[revised manuscript text omitted]
. Thus, the CG lightning peak currents were converted to signal strength with an arbitrary units (a. u.), using the method of Schütte et al. (1988), which measured the signal strength linearly depending on the distance. Schütte et al. (1987) concluded that the Weibull distribution can be used to estimate some characteristics related to thunderstorms.

15   Methodologically, in case of the weibull Weibull distribution of signal strength (Schutte et al. 1987), the signal acceptance of a sensor can be given by

$$
A(r) = \begin{cases} 0 & r < cr_0/s_{max} \\ 1 - \exp[-(\frac{s_{max}r/r_0 - c}{a})^b] & cr_0/s_{max} < r \leq cr_0/s_{min} \\ \exp[-(\frac{s_{min}r/r_0 - c}{a})^b] - \exp[-(\frac{s_{max}r/r_0 - c}{a})^b] & r > cr_0/s_{min} \end{cases}, \tag{14}
$$

where $s_{min}$ and $s_{max}$ are the lower and upper signal threshold, which will be 20 and 600 arbitrary units (a. u.), respectively; $r_0$ the standard distance, which will be 100 km; $r$ the distance to the sensor; $a, b,$ and $c$ are the scale, the shape and the location

20   parameter of the Weibull distribution of signal strength (Schütte et al. 1987, 1988).

Only two ADTD IMPACT sensors reporting a stroke are required to get a valid solution. Thus, the DE on a grid-cell can be determined as (Naccarato and Pinto 2009)

$$
A = A_1(r_1) \times A_2(r_2) \quad A_i \, ranked \qquad A_1(r_1) \geq A_2(r_2) \geq A_3(r_3) \geq \ldots \tag{15}
$$

where $A_i(r_i)$ denotes the acceptance of one sensor; $r_i$ (i=1,2,3,…) is the distance of the $i$th nearest sensor to the grid-cell

25   center and $A$ the grid-cell DE of the network.

Then the CG stroke density of each gridcell can be corrected using Eq. (2). After corrected the correction for DEsusing these deduced DEs (see Fig. 12), the CG stroke densities in the northeast mountains, metropolitan areas, south plains and southwest mountains increased significantly in comparison with the uncorrected CG stroke densities (see Fig. 3b4b-c). The corrected densities in metropolitan areas are mainly abovebetween 4-64 stroke/yr.km$^2$, which is higher than expected.

30   However, the relatively high CG stroke density remains in the plains. It is advisable that the network should be upgraded for

improvement of the network DE and  location accuracy. Maybe this anomaly can be explained using observed evidence.

Meanwhile, we calculated the strike-point density which is recommendable for lightning risk assessment (IEC62858) and it exhibits similar characteristics to that of corrected CG stroke density but a more sharply distribution (see Fig. 4d).

**5 Case study of lightning risk assessment in a resident sub-district**

[revised manuscript text omitted]

Cummins, K. L., and Murphy, J. M.: An overview of lightning locating systems: history, techniques, and data uses, with an in depth look at the US NLDN, IEEE T. Electromagn. C., 51: 499–518, 2009.

Curran, E. B., Holle, R. L., and Lopez, R. E.: Lightning casualties and damages in the United States from 1959 to 1994, J.

25  Climate, 13: 3448–3464, 2000.

Drüe, C., Hauf, T., Finke, U., Keyn, S., and Kreyer, O.: Comparison of a SAFIR lightning detection network in northern Germany to the operational BLIDS network. J. Geophy. Res., 112(D18114), doi:10.1029/2006JD007680, 2007.

Gabriel, K. R., and Changnon, S. A.: Temporal features in thunder days in the United States, Climatic Change, 15: 455–477, 1989.

30  Hu, H., Wang, J., and Pan, J.: The characteristics of lightning risk and zoning in Beijing simulated by a risk assessment model. Nat. Hazards Earth Syst. Sci.          , 14: 1985-2014. doi:10.5194/nhess-14-1985-2014, 2014.

Hu, H.: An Algorithm for Converting Weather Radar Data into GIS Polygons and its Application in Severe Weather Warning Systems. International Journal of Geography Information Science,      28(9): 1765-1780, 2014.

Hu, H: Spatiotemporal Characteristics of Rainstorm-Induced Hazards Modified by Urbanization in Beijing. J. Appl. Meteorol. Climatol., 54(7):1496-1509, 2015.

Holle, R. L., López, R. E., and Navarro, B. C.: Deaths, injuries, and damages from lightning in the United States in the 1890s in comparison with the 1990s, J. Appl. Meteorol., 44: 1563–1573, 2005.

Idone, V.P., Davis, D. A., Moore, P. K., Wang, Y,, Henderson, R. W., Ries, M., and Jamason, P. F.: Performance evaluation of the U.S. National Lightning Detection Network in eastern New York; Part I: Detection efficiency. J. Geophys. Res., 103: 9045–9056, doi:10.1029/ 98JD00154, 1998.

IEC62305-2. The technical committee of the International Electrotechnical Commission: Protection against lightning, IEC, Geneva, Switzerland, 2010.

IEC62858. Lightning density based on lightning location systems (LLS)-General principles. IEC, Geneva, Switzerland, 2015.

Kaplan, S. and Garrick, B. J.: On the quantitative definition of risk, Risk Anal., 1: 11–27, 1981.

Kar, S. K., Liou, Y. A.: Enhancement of cloud-to-ground lightning activity over Taipei, Taiwan in relation to urbanization. Atmos. Res., 147–148: 111–120, 2014.

Krider, E. P., Noggle, R. C., Pifer, A. E., and Vance, D. L.: Lightning Direction-Finding Systems for Forest Fire Detection.[J]. B. Am. Meteorol. Soc.,, 61(61): 980-986, 1980.

López, R. E., Holle, R. L.: Changes in the number of lightning deaths in the United States during the twentieth century. J. Climate, 11: 2070–2077, 1998.

Mäkelä, A., Tuomi, T. J., and Haapalainen, J.: A decade of high‐latitude lightning location: Effects of the evolving location network in Finland. J. Geophy. Res., 115(D21124), doi:10.1029/2009JD012183, 2010.

Mazarakis, N., Kotroni, V., Lagouvardos, K., and Argiriou, A. A.: Storms and Lightning Activity in Greece during the Warm Periods of 2003–06. J. Appl. Meteorol., 47: 3089-3098, 2008.

Mills, B., Unrau, D., Pentelow, L., and Spring, K: Assessment of lightning-related damage and disruption in Canada. Nat. Hazards, 52: 481–499. DOI 10.1007/s11069-009-9391-2, 2010.

Naccarato, K. P., Pinto, Jr. O.: Improvements in the detection efficiency model for the Brazilian lightning detection network (BrasilDAT). Atmos. Res., 91: 546–563, 2009.

Petrov, N. I., and D'Alessandro, F.: Assessment of protection system positioning and models using observations of lightning strikes to structures, Proc. R. Soc. Lond. A, 458, 723–742. doi:10.1098/rspa.2001.0906, 2002.

Rakov, V. A., and Uman, M. A.: Some properties of negative cloud-to-ground lightning flashes versus stroke order. J. Geophys. Res., 95(D5): 5447– 5453, 1990.

Rizk, F. A. M.: Modelling of lightning incidence to tall structures, part II, application, IEEE T. Power Deliver., 9: 172–193, 1994.

Rose, L. S., Stallins, J. A., and Bentley, M. L.: Concurrent cloud-to-ground lightning and precipitation enhancement in the Atlanta, Georgia (United States), urban region, Earth Interact., 12, 1–30, doi:10.1175/2008EI265.1, 2008.

Rudlosky, S., and Fuelberg, H. E.: Pre- and Postupgrade Distributions of NLDN Reported Cloud-to-Ground Lightning

Characteristics in the Contiguous United States. Mon. Wea. Rev., 138, 3623-3633, 2010.

Saraiva, A. C. V., Saba, M. M. F., Pinto, Jr. O., Cummins, K.L., Krider, E. P., and Campos, L. Z. S. A comparative study of negative cloud‐to‐ground lightning characteristics in São Paulo (Brazil) and Arizona (United States) based on high‐speed video observations. J. Geophy. Res. 115(D11102), doi:10.1029/2009JD012604, 2010.

5    Schulz, W., Cummins, K. L., Diendorfer, G., and Dorninger, M.: Cloud-to-ground lightning in Austria: A 10-year study using data from a lightning location system. J. Geophy. Res., 110: D09101. doi:10.1029/2004JD005332, 2005.

Schütte, T., Salka, O., and Israelsson, S.: The use of the Weibull distribution for thunderstorm parameters. J. Climate and Applied Meteorology, 26: 457-463, 1987.

Schütte, T., Cooray, V., and Israelsson, S.: Recalculation of lightning location system acceptance using a refined damping

10   model. J. Atmos. Oceanic. Technol., 5: 375-380, 1988.

Shepherd, J. M., Pierce, H., and Negri, A. J. Rainfall modification by major urban areas: observations from spaceborne rain radar on the TRMM satellite, J. Appl. Meteorol., 41, 689–701, 2002.

Smith, K.: Environmental Hazards: Assessing Risk and Reducing Disaster, 2nd Edn., Routledge, New York, USA, 1996.

Stallins, J. A., Bentley, M. L., and Rose, L. S.: Cloud-to-ground flash patterns for Atlanta, Georgia (USA) from 1992 to 2003.

15   Climate Res., **30,** 99–112, 2006.

Stallins, J. A., and Rose, L. S.: Urban lightning: current research, methods, and the geographical perspective, Geography Compass, 2: 620–639, doi:10.1111/j.1749-8198.2008.00110.x, 2008.

Steiger, S. M., Orville, R. E., and Huffines, G.: Cloud-to-ground lightning characteristics over Houston, Texas: 1989–2000. J. Geophys. Res., **107,** 4117, doi:10.1029/2001JD001142, 2002.

20   Visacro, S., Vale, M. H. M., Correa, G., and Teixeira, A.: Early phase of lightning currents measured in a short tower associated with direct and nearby lightning strikes. J. Geophy. Res., 115(D16104), doi:10.1029/2010JD014097, 2010.

Vogt, B. J.: Exploring cloud-to-ground lightning earth highpoint attachment geography by peak current, Earth Interact., 15: 1–16, 2011.

Warner, T., Helsdon, Jr. J. H., Bunkers, M. J., Saba, M. M. F., and Orville, R. E.: Upward Lightning Triggering Study. Bull.

25   Amer. Meteor. Soc., 94(5): 631-635, 2013.

Wisdom, M. D.: Lightning fatalities in Swaziland: 2000–2007, Nat. Hazards, 50, 179–191, 2009.

Zhang, W., Meng, Q., Ma, M., and Zhang, Y.: Lightning casualties and damages in China from 1997 to 2009, Nat. Hazards, 57, 465–476, 2011.

Yao, W., Zhang, Y., Meng, Q., Wang, F., and Lu, W.: A Comparison of the Characteristics of Total and Cloud-to-Ground

30   Lightning Activities in Hailstorms. Acta Meteor. Sinica, 27(2): 282-293, 2012.

**Table** 1. Estimating terrain factor of a structure accounting for its surrounding topography (defined by IEC62305-2, 2010)

| Description of the surrounding topography | $C_d$ |
|---|---|
| Higher than the top of the structure | 0.25 |
| As high as the top of the structure | 0.5 |
| On flat ground | 1 |
| On the top of a hill | 2 |

**Table 2.** The structure types corresponding to the lightning protective capability in Beijing

| Structure type | GIS identity | Protection measures | $p_a$ |
|---|---|---|---|
| General building | 211 | Iron infra-structure and framework as a lead-in wire in structure. | $10^{-4}$ |
| general structure with basement | 21109 | *Same as above* | $10^{-4}$ |
| bunk house | 212 | Effective soil equipotentialization | $10^{-2}$ |
| bunk house with basement | 21209 | *Same as above* | $10^{-2}$ |
| bridge gallery | 218 | Electrical insulation of exposed down-conductor | $10^{-2}$ |
| Special house | 229 | Iron infra-structure and framework as a lead-in wire in structure | $10^{-4}$ |
| Special house with basement | 22909 | *Same as above* | $10^{-4}$ |
| Ruined house | 214 | No protection measures | 1 |
| Hut | 215 | *Same as above* | 1 |
| Public lavatory | 3551 | Electrical insulation of exposed down-conductor | $10^{-3}$ |

[Figure]

Figure 1 Histogram of peak current probability density of (a) –CG lightning, (b) +CG lightning and identified +IC lightning on account of their peak currents less than 15 kA and (c) total CG lightning.

[Figure]

Figure 1 2 Distribution of the estimated network DEs and ADTD sensors around the Beijing district (enclosed by scarlet lines). The SDEs in Beijing metropolitan areas (red lines) are almost all above 55% and are lower than those in the surrounding areas, whereas the peak SDE zones located to the east of the metropolitan areas possess a maximum SDE of 81.1%.

[Figure]

Figure 2 3 Sketch map illustrates three types of underlying ground area samples (i.e., earthed structure, outdoor area under structure canopy and open-field area) in a 5 m spaced grid displayed in GIS.

(a)

(b)

[Figure]

(c)

[Figure]

(d)

[Figure]

Figure 4 Distribution of (a) CG flash density (flash/ yr.km2), (b) CG stroke density (stroke/ yr.km$^2$), (c) corrected CG stroke density (stroke/yr.km$^2$), and (d) strike-point density(strike/ yr.km$^2$). For convenience, the same legends for contour and shading were used in the CG flash density, CG stroke density, corrected CG stroke density and strke-point density plots.

[Figure]

[Figure]

[Figure]

Figure 3 4 Distribution of (a) CG flash density (fl/yr.km²), (b) CG stroke density (stroke/yr.km²), and (c) corrected CG stroke density (stroke/yr.km²). For convenience, the same legends for contour and shading were used in the CG flash density, CG stroke density and corrected CG stroke density plots.

(a)

(b)

[Figure]

(c)

[Figure]

Figure 4 5 Sensitivity zones in the sub-district of Malianwa, in case of (a) not accounting for the terrain factor, (b) accounting for terrain factors, (c) and displayed in Google Earth; these zones correspond well with the distribution of underlying structures and topographical features. For example, point *A* in the mountainous areas exhibits a high sensitivity, *B* in the dense structure areas exhibits a lower sensitivity on account of lightning rod effects produced by nearby structures, and *C* in an open-field area exhibits a relatively high sensitivity. Interestingly, the sensitivity of point *D* at an open sports field is obviously higher than that of its surrounding densely built-up areas.

(a)

[Figure]

(b)

[Figure]

(c)

[Figure]

Fig<s>.</s>ure 6 Lightning risk assessment of (a) ground sensitivity to lightning, <s>and</s> (b) NDLE deduced using corrected CG stroke density, and (c) NDLE deduced using strike-point density (displayed in <s>Google Earth</s>satellite imagery) zones in the Beijing international airport.